# Towards Anytime Optical Flow Estimation with Event Cameras

**DOI:** 10.3390/s25103158

**Published:** 2025-05-17

**Authors:** Yaozu Ye, Hao Shi, Kailun Yang, Ze Wang, Xiaoting Yin, Lei Sun, Yaonan Wang, Kaiwei Wang

**Affiliations:** 1State Key Laboratory of Extreme Photonics and Instrumentation, College of Optical Science and Engineering, Zhejiang University, Hangzhou 310027, China; yaozuye@zju.edu.cn (Y.Y.); haoshi@zju.edu.cn (H.S.); wangze0527@zju.edu.cn (Z.W.); yinxiaoting@zju.edu.cn (X.Y.); leo_sun@zju.edu.cn (L.S.); 2National Engineering Research Center of Robot Visual Perception and Control Technology, School of Robotics, Hunan University, Changsha 410082, China; yaonan@hnu.edu.cn

**Keywords:** event-based, optical flow, deep learning

## Abstract

Event cameras respond to changes in log-brightness at the millisecond level, making them ideal for optical flow estimation. However, existing datasets from event cameras provide only low-frame-rate ground truth for optical flow, limiting the research potential of event-driven optical flow. To address this challenge, we introduce a low-latency event representation, *unified voxel grid (UVG)*, and propose *EVA-Flow*, an *EV*ent-based *A*nytime *Flow* estimation network to produce high-frame-rate event optical flow with only low-frame-rate optical flow ground truth for supervision. Furthermore, we propose *rectified flow warp loss (RFWL)* for the unsupervised assessment of intermediate optical flow. A comprehensive variety of experiments on MVSEC, DESC, and our EVA-FlowSet demonstrates that EVA-Flow achieves competitive performance, super-low-latency (5 ms), time-dense motion estimation (200 Hz), and strong generalization.

## 1. Introduction

Optical flow estimation is a fundamental task in computer vision that aims to calculate the motion vector of each pixel in two consecutive frames of images. The event camera [1] is a new type of bio-inspired sensor that only responds to changes in the brightness of the environment. Compared with the traditional frame-based camera, which integrates the brightness at a certain time interval (exposure time) and outputs an image, the event camera has no exposure time and responds once the log value of the pixel intensity changes beyond a certain threshold, and the response is microsecond-level and asynchronous. As optical flow estimation aims to estimate the motion of a scene which aligns with the event camera’s ability to detect changes, it makes the event camera a suitable choice for optical flow estimation [2,3]. Furthermore, event cameras offer distinct advantages for optical flow estimation, such as high temporal resolution and high dynamic range (>120 dB compared to about 60 dB of traditional frame-based cameras [1]). The high dynamic range capability of event cameras can solve the problem of under- or over-exposed images produced by traditional frame-based cameras in poor illumination conditions (such as at night) or high-dynamic scenes (such as cars entering or exiting tunnels) [4,5,6], which can cause inaccurate optical flow estimation in subsequent processing. The high temporal resolution of event cameras eliminates the problem of motion blur even in high-speed motion scenes. Additionally, the high temporal resolution output of event cameras provides hardware support for super high-frame-rate optical flow estimation.

However, most current event-based optical flow estimation methods fail to fully exploit the high temporal resolution and low-latency characteristics of events, resulting in low frame rates for the estimated optical flow. They typically transform the event stream over a time interval into a tensor represented as voxels [2], which are then fed into a frame-based optical flow estimation network similar to image-based approaches [2,3,7]. Moreover, the output frame rate is limited by the underlying dataset. The two most commonly used benchmarks for event-based optical flow estimation, MVSEC [8] and DSEC [9] have low frame rates for their ground-truth optical flow, with DSEC having a frame rate of 10 Hz and MVSEC having a frame rate of 20 Hz. While significant progress has been witnessed in the field, the limited temporal resolution greatly restricts the potential of event-based optical flow estimation in real-world applications.

We present EVA-Flow, a deep architecture designed for *EV*ent-based *A*nytime optical *Flow* estimation. EVA-Flow delivers numerous benefits, such as low latency, high frame rate, and high accuracy. Figure 1a–c illustrates the differences between previous event-based optical flow estimation and our proposed approach toward anytime optical flow estimation. While other methods are limited by the dataset’s frame rate, this framework achieves ultra-low latency and offers optical flow estimation at a frequency 20 times higher. Figure 1d illustrates the discrepancy between the predictions of this model and E-RAFT [3] using a sample from the test sequence of the DSEC dataset. The scene depicts a car making a turn inside a tunnel. Our EVA-Flow offers continuous trajectory tracking within a 100 ms time range of the input sample. The transition from yellow to red indicates varying point positions at different times in the trajectory of the car turning inside the tunnel. In contrast, E-RAFT is restricted to generating a single optical flow.

To achieve high-frame-rate event output, we first propose the *unified voxel grid (UVG)*, which generates a bin of UVG representation as soon as the events in a small time interval are ready. Our UVG representation achieves a higher frame rate up to a factor of *N* and a lower latency of 1/N compared to the voxel grid representation [7]. *N* is a hyperparameter that can be adjusted to meet the specific requirements of the task. These bins are then sequentially fed into our EVA-Flow, which consists of a multi-scale encoder and our stacked *spatiotemporal motion recurrent (SMR)* module. SMR is proposed to predict temporally dense optical flow and enhances the accuracy via spatial-temporal motion refinement. Notably, the architecture is specially designed for extreme *low-latency* optical flow estimation: as each event bin arrives, it is immediately processed through the SMR module, yielding the optical flow result for the current time instance. This architectural design eliminates the need to wait for all bins of the entire voxel grid to arrive in order to achieve low-latency and high-frame-rate optical flow output.

Furthermore, we introduce rectified flow warp loss (RFWL), a novel method for the precise, unsupervised evaluation of event-based optical flow precision, and apply it to assess the accuracy of our time-dense optical flow estimations.

By comparing our EVA-Flow with other event-based optical flow estimation methods on widely recognized public benchmarks, including the DSEC [9] and MVSEC [8] datasets, we have established that our approach achieves performance on par with leading methods, offering advantages such as reduced latency (5 ms) and higher frame rates (200 Hz). Additionally, the evaluation of time-dense optical flow on DSEC, MVSEC, and EVA-FlowSet underscores the reliability of our framework’s time-dense optical flow, even under the constraint of being solely supervised with low-frame-rate optical flow ground truth during training. Furthermore, the Zero-Shot results derived from both the MVSEC dataset and our EVA-FlowSet clearly illustrate the strong generalization capabilities of our method.

In summary, our key contributions are threefold:**EVA-Flow Framework**: An ***EV**ent-based **A**nytime optical **Flow** estimation framework* that achieves breakthroughs in latency and temporal resolution. The architecture features two novel components: (1) *unified voxel grid (UVG)* representation enabling ultra-low latency data encoding, and (2) A *time-dense feature warping* mechanism where shared-weight SMR modules propagate flow predictions across temporal scales. This structural design fundamentally enables *single-supervision learning*—only final outputs require low-frame-rate supervision while implicitly regularizing intermediate time steps through feature warping recursion.**Rectified Flow Warp Loss (RFWL)**: A new unsupervised metric specifically designed for evaluating event-based optical flow precision. This self-consistent measurement provides theoretical guarantees for temporal continuity validation of high-frequency optical flow estimation.**Systematic Validation**: Comprehensive benchmarking demonstrates *competitive accuracy*, *super-low latency*, *time-dense motion estimation* and *strong generalization capability*. Quantitative analyses using RFWL further confirm the reliability of our continuous-time motion estimation.

## 2. Related Work

### 2.1. Optical Flow Estimation

Recently, the field of optical flow estimation has witnessed remarkable advancements owing to the progress of deep learning. FlowNet [10] firstly introduces an end-to-end network with a U-Net architecture to estimate optical flow directly from image frames and many subsequent works [11,12,13] follow this architecture. Later, the works of [14,15] utilize a warp mechanism to conduct a multi-scale refinement, achieving higher accuracy with less computational costs. RAFT [16] proposes a recurrent optical flow network that uses all-pair correlations and GRU iterations to produce high-precision flow results through iterative refinement, which achieves a significant improvement in accuracy. Most of the later optical flow studies are, in essence, improvements upon RAFT. CSFlow [17] introduces a decoupled strip correlation layer to enhance the capacity to encode global context. FlowFormer [18] explores the self-attention mechanism in recurrent flow networks to achieve further flow accuracy boosts, albeit with larger parameter usage. In summary, the use of warp mechanism, correlation volume, and RNN with iterative refinement has brought about significant advancements in the field of optical flow, allowing for improved accuracy and more efficient computation.

### 2.2. Event-Based Optical Flow

Event optical flow estimation can be categorized into two methodologies: model-based and learning-based approaches. The model-based approach to event optical flow estimation encompasses two distinct strategies. The first strategy involves fitting a local spatiotemporal plane to the event point cloud and subsequently leveraging the derived plane fitting parameters to ascertain the optical flow. This methodology has been extensively explored in the literature, as evidenced by the works of Benosman et al. [19], Mueggler et al. [20], and Low et al. [21]. The second strategy is predicated on the contrast maximization framework, as articulated by Gallego et al. [22]. This approach entails the computation of optical flow by optimizing the contrast of a motion-compensated event frame, a process that has been further elucidated by Shiba et al. [23,24]. However, model-based event optical flow algorithms [19,20,21,25] generally require denoising algorithms like [26] for event preprocessing to improve the accuracy.

Thanks to the availability of large-scale event optical flow datasets [8,9], learning-based event optical flow estimation algorithms [2,3,7] have achieved superior accuracy compared to model-based algorithms, which is also more robust to event noise.

EV-FlowNet [7] firstly utilizes bilinear interpolation on discrete events in both spatial and temporal dimensions. This transformation converts sparse and continuous events into a voxel-based tensor called a voxel grid. Subsequently, the voxel grid is input to FlowNet [10] for estimating event optical flow. This paradigm has been adopted by numerous subsequent works [3,27,28,29,30]. E-RAFT [3], leveraging the sophisticated RAFT [16] optical flow estimation network, has markedly improved the precision of event-based optical flow estimation and has long been regarded as the state-of-the-art method on the DSEC-Flow dataset [9]. Recently, E-Flowformer [31] leveraged transformers [32] to enhance network performance, achieving higher optical flow estimation accuracy with an expanded synthetic dataset, BlinkFlow [31]. TMA [33] and IDNet [34] have leveraged the abundant motion data recorded by event cameras during intermediate time frames to improve optical flow estimation, further enhancing the precision of event-based optical flow. Although these methods achieve high-precision optical flow estimation, they fail to capitalize fully on the low latency and swift response capabilities of event cameras. Constrained by the dataset’s frame rate, they are restricted to generating optical flow estimates at similarly low frame rates, specifically 10 Hz on the DSEC-Flow dataset.

Event cameras offer brightness variation data with super-high temporal resolution; however, there is a paucity of methods for high-frame-rate event optical flow estimation. Due to the lack of time-continuous optical flow ground truth, several studies [35,36] have employed unsupervised contrast-maximization-based losses [23], resulting in lower accuracy for these methods. These self-supervised methods typically perform well on small optical flow datasets (MVSEC) but exhibit poor accuracy on datasets with large displacements (DSEC-Flow). Wan et al. [37] develop an event-image fusion approach that combines events and images for temporal feature learning. While effective for continuous scene understanding, this method requires additional image inputs and recomputing event embeddings for each step during feature fusion. Ponghiran et al. [38] utilized recurrent neural networks to estimate time-dense optical flow from events, but their method required sequential training and was dependent on high-frame-rate ground-truth optical flow for supervision. On the DSEC-Flow dataset, they employed interpolation to achieve high-frame-rate optical flow ground truth; however, due to the large displacements and the nonlinear nature of many motions within this dataset, their accuracy on DSEC-Flow was comparatively poor. Gehrig et al. [39] introduced a continuous-time event optical flow algorithm that leverages dense temporal correlation to estimate pixel motion trajectory parameters (Bézier curves) across an entire time interval, facilitating continuous optical flow estimation. However, this method estimates the pixel motion trajectory for a specified time period only after the accumulation of all events (100 ms on DSEC-Flow). In contrast, our novel approach, EVA-Flow, attains high accuracy, minimal data latency, a very high prediction frequency, and superior model generalization through comprehensive end-to-end training, necessitating only low-frame-rate optical flow ground truth for supervision.

## 3. Method

In this paper, we propose an ***EV**ent-based **A**nytime optical **Flow** estimation (EVA-Flow)* framework, as shown in Figure 2, which achieves high accuracy, time-dense flow estimation, and low-latency input and output with supervision solely dependent on low-frame-rate optical flow. We first put forward a *unified voxel grid (UVG)* representation in Section 3.1, which generates event representations with low latency. These UVG bins are sequentially fed into EVA-Flow, where high-frame-rate optical flow estimations are generated with low latency using a *Spatiotemporal Motion Recurrent (SMR)* module, which is detailed in Section 3.2. Then, the details of the loss function and the supervision regime are described in Section 3.3. Finally, we propose a new criterion to evaluate the reliability of intermediate high-frame-rate optical flow in Section 3.4.

### 3.1. Event Representation: Unified Voxel Grid

The event representation is crucial for the event-based model [40,41]. A voxel grid [7] is a commonly used event representation that utilizes the temporal dimension of event data, and its effectiveness for optical flow estimation tasks has been verified in many studies [3,7,34,39]. The principle of a voxel grid is to discretize the spatially sparse and temporally continuous information of events by averaging over time and then utilize bilinear interpolation in both spatial and temporal dimensions to obtain a tensor representation of the event data in a voxel form. The tensor has a dimension of *B* × *H* × *W*, where *B* represents the number of time steps, and its size determines the sampling accuracy of the event data in the temporal dimension. Each channel of the voxel grid can be viewed as an event representation at a specific time. Given a set of events ${\left\{\left(x_{i},y_{i},t_{i},p_{i}\right)\right\}}_{i\in [1,N]}$ and *B* bins to discretize the time dimension, the definition of voxel grid is depicted as follows:(1)$$\begin{array}{cc} \; & t_{i}^{*}=\left(B-1\right)\left(t_{i}-t_{1}\right)/\left(t_{N}-t_{1}\right),\\ \; & k_{b}\left(a\right)=\max\left(0,1-|a|\right),\\ \; & VG\left(x,y,t\right)=\sum_{i}p_{i}k_{b}\left(x-x_{i}\right)k_{b}\left(y-y_{i}\right)k_{b}\left(t-t_{i}^{*}\right), \end{array}$$
where $k_{b}\left(a\right)$ denotes the bilinear sampling kernel.

Continuous prediction of high-frame-rate optical flow necessitates corresponding high-frequency event representations. A critical limitation of the standard voxel grid, illustrated in Figure 3, is that generating the complete grid requires waiting for all events within the entire time interval *T*, making it inherently batch-based and unsuitable for low-latency applications. Furthermore, the effective temporal support for the first (b=0) and last (b=B−1) bins is often halved, leading to representational inconsistency.

To address these limitations, particularly the latency issue, we propose the *unified voxel grid (UVG)*. UVG enables sequential, low-latency event representation. Instead of waiting for the full duration *T*, UVG allows for the computation of an event representation centered around a specific time $t_{current}$ using only events within a localized temporal window. We achieve this by defining each bin $Bin_{b}$ consistently with a fixed temporal support 2τ, centered at its timestamp $t_{b}$. The representation for a bin is calculated using events $e_{i}$ falling within $t_{b}-\tau <t_{i}<t_{b}+\tau$:(2)$$\begin{array}{cc} \; & Bin_{b}^{UVG}\left(x,y\right)=\sum_{i}p_{i}k_{b}\left(x-x_{i}\right)k_{b}\left(y-y_{i}\right)k_{b}\left(\frac{t-t_{b}}{\tau }\right) \end{array}$$

This design makes it possible to obtain the current representation of time for each τ (5 ms for 21 channels with unified voxel grid on DESC), while the voxel grid must wait for the entire period of *T* (100 ms on DSEC) to generate a complete event representation. This low-latency input can be coupled with our low-latency time-dense optical flow estimation architecture for tracking the optical flow at each τ time step as it arrives which is essential for high-frame-rate optical flow estimation.

### 3.2. Event Anytime Flow Estimation Framework

In this subsection, we propose the EVA-Flow framework, which predicts time-dense optical flow from serial event bins with low latency. The architecture of EVA-Flow is illustrated in Figure 2. The raw events are converted to UVG and then sequentially fed into the encoder, bin by bin, to generate a four-level feature pyramid ($f_{i}^{1}\;\to \;f_{4}^{N}$). The feature pyramid is fed into our stacked *spatiotemporal motion recurrent (SMR)* modules to update the optical flow in both temporal and spatial dimensions while outputting the refined optical flow results from $t_{0}$ to the current bin.

**Serialized low-latency input and output.** The UVG inputs are sequentially fed into the network, which contributes to the low latency at the data entry level. During the inference phase, after entering an event bin (excluding the first bin used for initialization), the current optical flow can be directly predicted using the encoder and the stacked SMR module. During the training phase, to facilitate end-to-end training, we follow the frame rate of the dataset and input a full period of UVG at once as a training sample. Following the UVG input of N, B, H, W, the UVGs of different bins (i.e., with different channels) are converted to the batch dimension and concatenated (resulting in a dimension of N×B, H, W) before being passed through the encoder. Once the feature maps are obtained, the corresponding feature maps of different bins are sequentially input into the stacked SMR module, which outputs time-dense optical flow ($F_{0,1}\to F_{0,B-1}$).

**Spatiotemporal Motion Recurrent (SMR) module.** To achieve accurate and time-dense optical flow estimation, we propose the spatiotemporal motion recurrent (SMR) module, illustrated in Figure 2. The core idea of SMR is to employ a hierarchically stacked structure for coarse-to-fine spatial refinement combined with ConvGRU-based temporal recurrence. This design inherently promotes stable convergence and high accuracy. Specifically, the coarse-to-fine spatial processing provides robust initial estimates and guides the refinement process across scales, simplifying the overall estimation task. Concurrently, the iterative residual updates within each level encourage gradual convergence by successively correcting the flow estimate. Moreover, the ConvGRU architecture [42], with its gating mechanisms, ensures stable learning and propagation of temporal motion information, effectively capturing motion dynamics over time. The process within each SMR unit is governed by the following:(3)$$\begin{array}{cc} \; & \hat{f_{j}^{i}}=\mathrm{Warp}\left(f_{j}^{i},F_{0,j}^{i-1}\right)\\ \; & Ht_{j}^{i}=\mathrm{ConvGRU}(Ht_{j-1}^{i},concat\left(\hat{f_{j}^{i}},\mathrm{Up}\left(F_{0,j}^{i-1}\right),\mathrm{Up}\left(Ht_{j}^{i-1}\right)\right)\\ \; & F_{0,j}^{i}=F_{0,j}^{i-1}+\mathrm{FlowHead}\left(Ht_{j}^{i}\right) \end{array}$$
where the superscript *i* denotes the feature pyramid level (from coarse to fine), and the subscript *j* indicates the temporal processing step.

Structurally, the SMR module operates along two primary dimensions:

**Vertical Dimension (Spatial Coarse-to-Fine Refinement):** At any given time step *j*, the SMR modules are stacked vertically across different spatial resolutions (pyramid levels *i*). Information propagates from coarser levels (i−1) to finer levels (*i*). Specifically, the input features $f_{j}^{i}$ at level *i* are warped using the upsampled flow estimate $F_{0,j}^{i-1}$ from the coarser level. This aligned feature $\hat{f_{j}^{i}}$, along with the upsampled flow and the upsampled hidden state $Ht_{j}^{i-1}$, are fed into the ConvGRU unit. The resulting hidden state $Ht_{j}^{i}$ is then processed by a flow head (implemented with two convolutional layers) to predict a residual flow, which updates the coarser flow estimate $F_{0,j}^{i-1}$ to yield the refined flow $F_{0,j}^{i}$ at the current level. This hierarchical refinement progressively enhances spatial accuracy. Importantly, the optical flow $F_{0,j}^{i}$ for the current time step *j* is predicted using only the features $f_{j}^{i}$ and state information available up to that time step. This enables low-latency flow estimation, as the output for time *j* can be generated as soon as the corresponding event features are processed.

**Horizontal Dimension (Temporal Recurrence):** Within each spatial level *i*, the SMR module processes information sequentially across time steps *j*. The ConvGRU [42] plays a crucial role here, utilizing its internal hidden state ($Ht_{j-1}^{i}$ from the previous time step) to update the current hidden state $Ht_{j}^{i}$. This recurrence mechanism maintains temporal consistency and allows the module to model motion dynamics over time. SMR modules within the same spatial level *i* (horizontally) share weights across different time steps *j*, promoting efficient learning of temporal patterns. Conversely, SMR modules at different spatial levels *i* (vertically) use distinct weights to accommodate varying feature resolutions and channel dimensions.

**Time-dense feature warping.** Unlike other methods [33,34] that rely on the assumption of linear motion between two ground-truth flows, our approach can handle non-linear motion throughout the entire UVG period. As depicted in Figure 2, during each warping iteration, the dense temporal optical flow from the preceding level is employed to warp the features of the current level at each time step. This structure guarantees consistent prediction patterns of optical flow for each resolution level and time step (SMR modules at the same level are using shared weights), implicitly providing SMR modules with the ability to predict optical flow at the current time step. By implementing this approach, we only need to supervise the final output $F_{0,B-1}$ for implicit monitoring of optical flow at intermediate time steps.

### 3.3. Supervision

Following E-RAFT [3], we also used L1 loss as the loss function. Specifically, we calculate the L1 distance between the ground-truth optical flow and the last optical flow prediction $F_{0,B-1}$ of our model as the final loss function. The loss function is defined as follows:(4)$$Loss=\left|\right|F_{gt}-F_{pre}{\left|\right|}_{1}.$$

Due to network structure design (see explanation in Section 3.2 in **Time-dense feature warping**), we do not need to supervise the optical flow prediction at intermediate time steps.

### 3.4. Rectified Flow Warp Loss

Motion compensation, as described in [22], entails aligning all events to a reference time by utilizing the estimated event optical flow. Specifically, for each event, the flow value at the event’s position and the time difference from the event’s occurrence to the reference time are used to determine the event’s position at the reference time (see Equation (Equation 5)). Henceforth, we refer to the motion-compensated event count image as the Image of Warped Events (IWE).

Given events $E=({\left\{\left(x_{i},y_{i},t_{i},p_{i}\right)\right\}}_{i\in [1,N]})$, per-pixel flow estimation F and a reference time $t^{\prime }$, the IWE is defined as follows:(5)$$I\left(E,F\right)=\left(\begin{array}{c} x_{i}^{\prime }\\ y_{i}^{\prime } \end{array}\right)=\left(\begin{array}{c} x_{i}\\ y_{i} \end{array}\right)+\left(t^{\prime }-t_{i}\right)F\left(x_{i},y_{i}\right).$$

Accurate flow estimation ensures that events generated by the same edge are aligned to the same pixel location, resulting in higher contrast in the IWE than in the original event frame. Stoffregen et al. [43] utilized this principle to propose flow warp loss (FWL), which can assess the accuracy of event optical flow estimation in an unsupervised way.(6)$$\mathrm{FWL}:=\frac{\sigma^{2}\left(I\left(E,F\right)\right)}{\sigma^{2}\left(I\left(E,0\right)\right)}$$

In theory, a more accurate estimation of optical flow results in a clearer, higher contrast, and larger variance for the image after motion compensation. This is because events occurring at the same location in the scene are aligned to the same point. Therefore, a more accurate prediction of optical flow leads to a larger FWL. In general, the image tends to become sharper after motion compensation, indicating an FWL value greater than 1. However, in practice, we find that the FWL can be less than 1, yet the IWE remains visibly sharper than the original, as depicted in Figure 4. This phenomenon, where FWL of IWE is less than 1, has also been observed in the study by Shiba et al. [23]. Upon analysis, we determined that motion compensation has caused some events to be warped outside the image boundaries, leading to a reduced event count in the IWE compared to the original event frame. Thus, we propose *rectified flow warp loss (RFWL)*, which normalizes image brightness based on the total event count (i.e., the sum of pixel values) in the event frames before and after motion compensation. The formulation is given as follows:(7)$$\mathrm{RFWL}:=\sigma^{2}\left(\frac{I(E,F)}{\sum I(E,F)}\right)/\sigma^{2}\left(\frac{I(E,0)}{\sum I(E,0)}\right).$$

As depicted in Figure 4, the FWL fails to accurately assess the precision of optical flow estimation. The IWEs in columns 1, 2, and 4 display higher sharpness than the original event frames, yet the corresponding FWL values are less than 1. Notably, the IWE in column 3, which exhibits a markedly sharper resolution than the original, has an FWL value of 1.12. In contrast, our proposed rectified flow warp loss can accurately reflect the relative sharpness of IWEs compared to the original event frames.

## 4. Experimental Results

Our model underwent evaluation on the DSEC [9], MVSEC [2], and self-collected EVA-FlowSet. Section 4.1 offers an introduction to the aforementioned datasets. Section 4.2 presents the implementation details of our approach. Section 4.3 demonstrates the evaluation of our model using the regular flow evaluation prototype. Section 4.4 reveals the evaluation results of our model for time-dense optical flow. Section 4.5 encompasses ablation studies.

### 4.1. Datasets

**DSEC.** The DSEC dataset, introduced by Gehrig et al. [9], consists of 24 sequences captured in real-world outdoor driving scenes. The dataset includes a total of 7800 training samples and 2100 testing samples, captured at a resolution of 640 × 480. It covers both daytime and nighttime scenarios, encompassing small and large optical flows. Moreover, the DSEC dataset offers high-precision sparse ground-truth values for optical flow. As the dataset does not include an official validation set, we divided the training dataset randomly using a fixed seed. To create the training and validation sets, we allocated them in a 4:1 ratio. Consequently, only 80% of the training data was utilized for training our model.

**MVSEC.** MVSEC [2] is a classic real-world event optical flow dataset that comprises sequences from various indoor and outdoor scenes with a resolution of 346 × 260. The event density of the dataset is relatively sparse, and the optical flow ground-truth distribution mostly highlights small flows. As a result, two different time intervals, i.e., dt = 1 and dt = 4, are used for accuracy evaluation. dt = 1 means that adjacent grayscale frames are used as a sample, whereas dt = 4 represents a four-fold increase in the grayscale frame interval. We follow the E-RAFT convention [3], conduct training solely on the outdoor day 2 sequence, and evaluate on 800 samples data from outdoor day 1.

**EVA-FlowSet.** We collect and present EVA-FlowSet, a real-world dataset to assess the generalization and time-dense optical flow of our model. For data collection, we utilized a Davis-346 camera (manufacturer: iniVation AG, Zurich, Switzerland),  which is the same type used in the MVSEC dataset. The dataset consists of four sequences in total, with two sequences depicting fast-moving scenes and the other two showcasing regular motion scenes. During the data-collection process, the camera remains stationary, capturing the movement of a checkerboard calibration board as it travels along a curved trajectory at different predetermined speeds.

### 4.2. Implementation Details

For the DSEC dataset [9], we use an Adam optimizer with a batch size of 6 and a learning rate of 5 × $10^{-4}$ to train for 100k iterations on the training set we divided, which accounts for 80% of the total dataset. Then, we reduce the learning rate by a factor of 10 and continue training for another 10k iterations. During the model training on the DSEC dataset, two online data augmentation methods, namely random cropping and horizontal flipping, were employed. Random cropping was performed with a size of 288 × 384, while horizontal flipping was applied with a probability of 50%.

For the MVSEC dataset [2], we employ an Adam optimizer with a batch size of 3 and a learning rate of 1 × $10^{-3}$ to train for 120k iterations. We utilize random cropping with a size of 256 × 256 and horizontal flipping with a probability of 50% during training. For MVSEC scenes with dt = 1, we use the setting with #bins = 3; for MVSEC with dt = 4, we use the setting with #bins = 9.

### 4.3. Regular Flow Evaluation Prototype

**Evaluation on DSEC.** Table 1 presents a comprehensive evaluation on the DSEC-Flow benchmark [9], including performance and computational complexity metrics. The metrics assess accuracy via EPE (L2 endpoint error in pixels), AE (angular error in degrees), and nPE (percentage of pixels with optical flow magnitude error greater than n pixels); model complexity via Params (millions of parameters); and computational cost via GMACs (Giga Multiply-Accumulate operations per inference, in billions).

Analyzing the results, our EVA-Flow establishes itself as a leading solution for time-dense optical flow. Compared to other approaches capable of high-frequency output (Taming_CM, LSTM-FlowNet), EVA-Flow achieves superior temporal characteristics with the lowest input latency (5 ms) and the highest prediction rate (200 Hz). While delivering this enhanced temporal resolution, it maintains a competitive level of accuracy. Although slightly less accurate than the top-performing non-time-dense methods, EVA-Flow demonstrates an excellent trade-off between high-frequency prediction capabilities and flow estimation precision. Crucially, EVA-Flow achieves this performance with remarkable computational efficiency, making it highly suitable for resource-constrained applications.

Importantly, the reported GMACs reflect the cost for generating a single optical flow prediction. This allows a fair comparison of per-prediction efficiency, irrespective of the output rate (e.g., our 200 Hz vs. 10/100 Hz for others). EVA-Flow’s low per-prediction cost is what enables its high-frequency operation.

E-RAFT [3], a representative method for event-based optical flow, incorporates the RAFT’s [16] superior design features, such as correlation computation and iterative optical flow estimation, resulting in enhanced optical flow accuracy. It serves as the foundation for the latest state-of-the-art methods, TMA [33] and BFlow [39]. Consequently, we have chosen E-RAFT as a benchmark for our qualitative experiments.

The qualitative results of our method on the DSEC flow dataset are shown in Figure 5. To make a more intuitive comparison of the accuracy of optical flow estimation, we also visualized the IWEs. It can be seen from the regions of interest (ROIs) in Figure 5 that, compared with E-RAFT, our model can better distinguish the outline of small objects (such as poles) and obtain more accurate optical flow estimates. Our analysis suggests that the distinguishing feature of our approach, in contrast to E-RAFT, lies in the implicit use of warp-aligned event edges to extract motion features, whereas E-RAFT relies on a correlation volume. For small objects with simple, well-defined contours, warping and aligning event features is relatively straightforward. However, due to limited textures, the correlation-based method, which depends on measuring feature similarity, struggles to capture high-quality motion features in areas with insufficient texture, leading to the inferior performance of E-RAFT in such cases.

**Evaluation on the MVSEC dataset.** The evaluation results of the MVSEC benchmark [2] are presented in Table 2. SSL, USL, and SL stand for self-supervised-, unsupervised-, and supervised learning, respectively. While our model’s accuracy is slightly lower than that of E-RAFT based on the data in the table, it is worth highlighting that our zero-shot results display a significantly better accuracy when directly testing the performance of the DSEC-trained model on the MVSEC dataset. Especially noteworthy is our model’s performance at dt = 4, where our model’s EPE is halved compared to that of E-RAFT (0.96 vs. 1.93). These results demonstrate the superior generalizability of the proposed EVA-Flow.

The qualitative results of testing MVSEC under the zero-shot setting, i.e., the model is only trained on DSEC, can be found in Figure 6. Owing to the small optical flow of MVSEC, the motion compensation effect is indistinct under the dt = 1 setting. Hence, we apply a longer time range of grayscale frames, i.e., dt=4. There is a marked discrepancy in motion speed between the MVSEC and DSEC datasets, as depicted in Figure 5 and Figure 6. Nonetheless, our model is capable of achieving near-ground-truth optical flow results on the MVSEC dataset, while E-RAFT shows inadequate generalization performances, particularly in non-event ground areas. Additionally, our model demonstrates superior capabilities for small targets such as poles and road markings, accurately depicting their contours with improved motion compensation results.

**Evaluation on EVA-FlowSet.** Qualitative results of the curve motion scene dataset EVA-FlowSet can be found in Figure 7. By visualizing the optical flow images, it can be observed that, for both fast-moving scenes and normal scenes, our model can estimate the contour of moving objects more accurately than E-RAFT. The magnified ROI area shows that our model obtains sharper motion compensation results in the first three rows of fast-moving scenes, while E-RAFT’s motion compensation performance is comparable to our model in the last three rows of moderate-moving-speed scenes. E-RAFT calculates the correlation between the event voxel grid before and after the start time to estimate the optical flow, which implicitly assumes that the optical flow is constant throughout the entire time. In contrast, our method uses the SMR module iteratively in each time step to obtain continuous optical flow estimation, which is suitable for curve motion scenes. In scenes with moderate movement speeds, even if the motion trajectory is curved, the motion can be approximated as linear due to the relatively slow speed. Therefore, in cases of moderate movement speeds, the motion compensation results of E-RAFT are comparable to those of our model. However, in fast-moving scenes, the linear motion assumption of E-RAFT becomes invalid due to the inherently curved nature of the motion trajectory. In contrast, our model, which employs a dense optical flow estimation framework independent of the linear motion assumption, significantly outperforms E-RAFT in fast-moving scenarios.

### 4.4. Time-Dense Optical Flow Evaluation

With #bins = 15, our model produces 14 consecutive optical flow estimates from events within a 100 ms window on the DSEC test set. The output $F_{0,i}$ covers 0 ms to 100 × $\frac{i}{14}$ ms, while E-RAFT provides only a single estimate for the entire 0 ms to 100 ms range. Due to the absence of time-dense ground truth in the DSEC dataset, a direct comparison with E-RAFT is not feasible. To address this, we generate images of warped events (IWEs) using our model’s time-dense flow estimates and E-RAFT’s interpolated flow and evaluate them with RFWL (refer to Section 3.4). This process is illustrated in Figure 8. All DSEC test sequences are evaluated, and detailed results are presented in Table 3.

Since $F_{0,i}$ represents forward optical flow from 0 ms to 100 × $\frac{i}{14}$ ms, we use this to warp the events within that period to generate the corresponding IWE. Our model’s optical flow estimates at intermediate time steps are generally more accurate than E-RAFT’s interpolated results, leading to better RFWL performance. However, as *i* approaches the final time step (e.g., *i* = 14), E-RAFT’s more accurate final optical flow results in a higher RFWL. Nevertheless, across seven sequences, our method surpasses E-RAFT in average RFWL on five sequences, demonstrating that our approach reliably estimates high-frame-rate optical flow even without supervision from high-frame-rate ground truth.

We conducted a comparative analysis using the EVA-FlowSet, which comprises sequences featuring curved movements. The results, depicted in Figure 9, indicate a significant performance improvement of our approach compared to E-RAFT. Since the EVA-FlowSet lacks ground-truth optical flow, both E-RAFT and our model are only trained on the DSEC dataset. Figure 9 presents a visual representation of the evaluation results comparing our method with E-RAFT in real-world scenarios with varying motion speeds. Specifically, in two fast-moving sequences, our model demonstrates a notably superior intermediate RFWL compared to E-RAFT. We observe that the overall accuracy of EVA-Flow is significantly higher during periods of rapid motion, yet it experiences a slight decrease in accuracy towards the end of the event bin cycle. This phenomenon correlates with the aforementioned results of the DSEC dense-time validation experiments. In the other two scenarios characterized by normal-speed sequences, our model’s RFWL performance is comparable to that of E-RAFT. This equivalence arises because, at insufficient movement speeds along a curved path, the motion can be briefly approximated as linear.

We also assessed the efficacy of our dense temporal optical flow methodology from an alternative perspective. In the DSEC-Flow, where the ground-truth flow frame rate is 10 Hz and each sample spans 100 ms, we created a 200 ms ground truth for event optical flow by warping together the preceding and subsequent 100 ms optical flows of our validation set. Thereafter, we applied a model trained on the original dataset, which consists of 100 ms samples, to estimate this extended validation set. Table 4 (right side) displays the outcomes of this experiment. For the EVA-Flow model, which was trained with Bins set to *b*, we used the adjusted setting of $b^{\prime }=1+\left(b-1\right)\times 2$ for estimating the 200 ms optical flow. This approach maintains the duration of each bin as it was during training and aligns the bin lengths with the extended overall estimation period. Conversely, E-RAFT, when processing a 200 ms sample, adheres to its original configuration of 15 bins. The data in Table 4 show that our model’s error did not escalate when estimating the validation set with an extended time span; rather, its precision improved, with the average end point error (EPE) being less than double that of the test set with 100 ms samples. In comparison, E-RAFT’s EPE surpassed three times the EPE of the 100 ms sample test set.

We also observed similar results on the MVSEC dataset. Table 2 displays the zero-shot results for E-RAFT and our model, which was trained on DSEC and evaluated on MVSEC with #bins = 15. When assessing optical flow estimation on the MVSEC dataset with our model, we use the identical checkpoint but adjust the bin count according to varying dt values. Due to the difference in motion speeds between the slow-motion MVSEC and the fast-motion DSEC datasets, we set #bins = 4 for dt = 1 and #bins = 13 for dt = 4. It is apparent that our model achieves a lower endpoint error (EPE) of 0.96 at dt = 4, which is less than four times the EPE at dt = 1 (0.39).

Interestingly, the results indicate that as we progressively iterate and generate optical flow for longer time intervals, the average per-unit-time optical flow error does not increase but rather decreases. These findings provide clear evidence for the efficacy of our framework in estimating time-dense optical flow. Furthermore, it underscores a significant advantage of the framework’s adaptable nature: the flexibility to adjust the number of prediction bins. This adjustment minimizes the difference in event rates per bin between the training and prediction phases, thereby enhancing accuracy during deployment.

### 4.5. Ablation Study

**Unified voxel Grid:** Table 5 compares the results of our framework on the DSEC test set using a voxel grid [7] and our proposed unified voxel grid representation. The EPE of our proposed unified voxel grid is 7.3% lower than that of the voxel grid. This difference stems from the fact that each event bin in the unified voxel grid adheres to a consistent event representation, while the representation patterns of the first and last bins in the voxel grid do not match those of the middle bins (refer to Figure 3).

**Number of Bins (N) Setting:** The number of bins *N* in the UVG representation is a crucial hyperparameter that dictates the temporal granularity (τ=T/(N−1), where *T* is the total time window, e.g., 100 ms) and the maximum prediction frequency. Setting *N* involves a trade-off: higher *N* allows for finer temporal resolution and potentially higher prediction frequency but requires sufficient event density within each shorter interval τ.

*Impact During Training (DSEC):* We first investigated the effect of *N* during training on the DSEC dataset, evaluating the final End-Point Error (EPE). As shown in Table 4 (left), increasing *N* from 6 up to 21 leads to a consistent decrease in the final EPE. This improvement can be attributed to the network estimating smaller displacements in each recurrent step, simplifying the estimation task per iteration. However, when *N* increases further (e.g., to 31), the accuracy declines as individual bins start to contain insufficient event information, hindering robust flow estimation. Based on these results, we selected N=21 for our primary model trained on DSEC, as it provides a good balance, achieving a 20-fold increase in potential output frequency compared to standard methods while maintaining high final accuracy.

*Impact During Inference (EVA-FlowSet [T=20 ms] and Guidance):* While training with N=21 yields strong results for the *final* flow prediction, the optimal *N* for achieving the best *time-dense* flow quality during inference can vary depending on scene dynamics. To explore this, we conducted experiments on our EVA-FlowSet, evaluating the average time-dense RFWL using different inference values of *N* (from 2 to 21) on sequences with varying speeds, using the model trained with N=15. The results, presented in Figure 10, reveal that the optimal inference *N* indeed depends on motion speed. For high-speed sequences, peak RFWL (best time-dense quality) is achieved around *N* = 11–12 (τ≈2 ms), suggesting that fast motion generates enough events for reliable estimation even with short integration times. Conversely, for normal-speed sequences, performance peaks around *N* = 9 (τ=2.5 ms), indicating that slightly longer integration times are beneficial when motion is slower.

This analysis highlights the flexibility of EVA-Flow. While trained with a specific *N* (e.g., 21), the inference can be adapted. We provide the following guidance for practical application (for our EVA-Set):For high-speed scenarios, using τ≈2 ms during inference is recommended for optimal time-dense performance.For moderate/low-speed or low-light (sparse event) scenarios, using a longer τ>2.5 ms during inference is likely beneficial to ensure sufficient event accumulation per bin.

It is important to note that these ranges serve as initial recommendations. The optimal setting for τ (and consequently *N*) depends strongly on the specific characteristics of the scene (motion speed, light condition). This adaptability allows users to tune the inference granularity for the specific scenario, maximizing the quality of the time-dense optical flow output.

## 5. Limitations and Future Work

While EVA-Flow demonstrates significant advantages in low-latency, high-frequency optical flow estimation with high computational efficiency, we acknowledge several limitations that suggest avenues for future research:Accuracy Relative to Non-Time-Dense Methods: Although competitive, EVA-Flow’s final prediction accuracy (e.g., EPE) is slightly lower than some state-of-the-art non-time-dense methods that utilize computationally intensive correlation volumes and focus solely on maximizing accuracy between two distant time points. Our current SMR module relies on implicit warp alignment for efficiency and time-density. Future work could explore hybrid approaches, potentially incorporating lightweight correlation features or attention mechanisms to enhance accuracy, especially for complex motions, without drastically increasing latency or computational cost.Sparse Temporal Supervision: EVA-Flow achieves time-dense prediction but is still primarily supervised using ground-truth optical flow provided at a much lower frame rate (e.g., 10 Hz on DSEC). While our unsupervised RFWL metric helps validate intermediate flows, the network lacks direct, dense temporal supervision during training. Developing techniques for generating reliable time-continuous ground truth, perhaps through advanced interpolation or simulation, or exploring unsupervised/self-supervised learning objectives specifically designed for time-dense event flow could further enhance intermediate flow accuracy and reliability.Event-Only Input: The current EVA-Flow framework operates solely on event data. While this highlights the richness of information within event streams, incorporating asynchronous image frames from the event camera (like Davis sensors) could provide complementary information, particularly in static scenes, low-texture areas, or during periods of low event activity. Designing efficient multi-modal fusion architectures that leverage both event dynamics and frame appearance is a promising direction for improving robustness and overall accuracy.

## 6. Conclusions

In this paper, we propose EVA-Flow, a learning-based model for anytime flow estimation based on event cameras. Leveraging the unified voxel grid representation, our network is able to estimate dense motion fields bin-by-bin between two temporal slices. By employing a novel spatiotemporal motion refinement module for implicit warp alignment, the proposed EVA-Flow overcomes the low-temporal-frequency issue of event-based optical flow datasets, achieving high-speed and high-frequency flow estimation beyond low-frequency supervised signals. Our approach significantly outperforms other time-dense event flow methods in temporal capabilities and demonstrates exceptional computational efficiency (16.8 GMACs per prediction). It also sustains a competitive level of accuracy compared to non-time-dense techniques. Furthermore, we introduced rectified flow warp loss (RFWL), an unsupervised metric for evaluating time-dense predictions, and showed that EVA-Flow’s prediction granularity can be flexibly adjusted during inference to suit varying scene dynamics. The combination of high temporal resolution, low latency, competitive accuracy, high computational efficiency, and adaptability makes EVA-Flow a highly practical and promising solution for real-time motion perception tasks on resource-aware platforms.

## Figures and Tables

**Figure 1 sensors-25-03158-f001:**
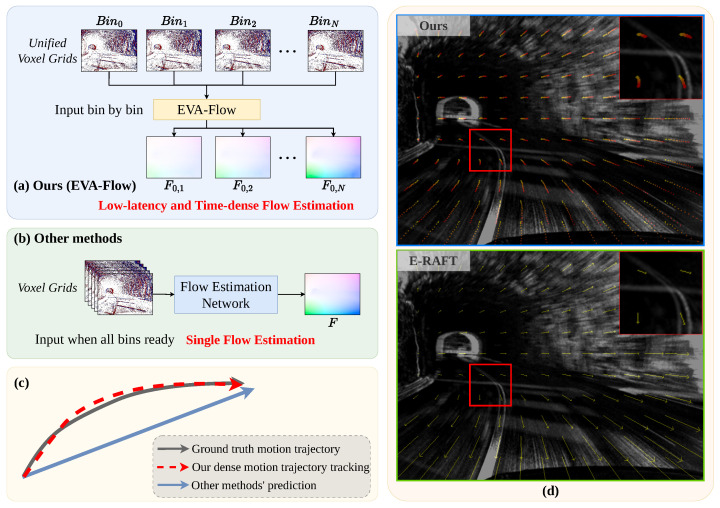
(**a**,**b**) EVA-Flow vs. previous event-based optical flow networks. (**c**) Comparison between our dense motion trajectory and other methods. (**d**) EVA-Flow’s dense motion trajectory vs. E-RAFT [3] on the DSEC dataset [9].

**Figure 2 sensors-25-03158-f002:**
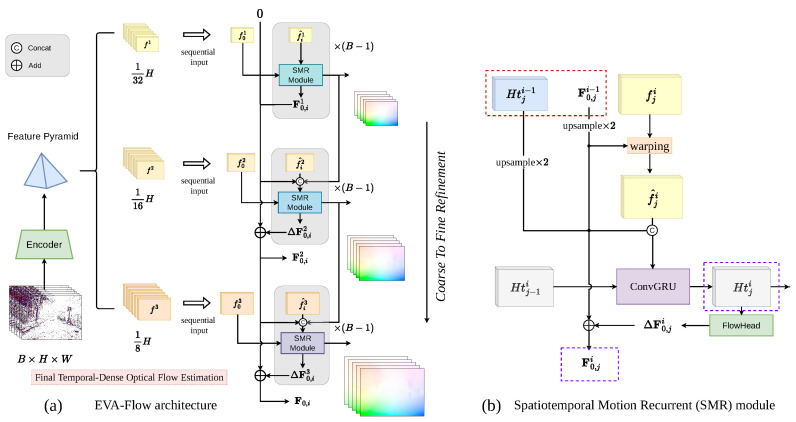
Architecture of our *Event Anytime Flow Estimation (EVA-Flow)* framework (**a**) and SMR module (**b**). $f^{i}$ denotes the features of the *i*-th level in a feature pyramid, while $f_{j}^{i}$ denotes the *j*-th event bin’s feature in $j^{i}$. $F_{0,i}$ denotes flow prediction from time $t_{0}$ to $t_{i}$. The red dashed box indicates the previous layer’s SMR module output, while the purple box shows the current layer’s output, destined for the next layer’s SMR module.

**Figure 3 sensors-25-03158-f003:**
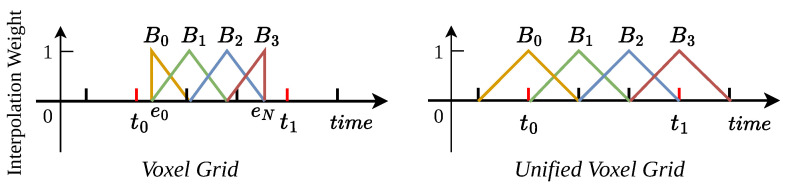
The proposed *Unified voxel grid* compared to voxel grid [7]. Different colored lines demonstrate the time range of events used for interpolation in different bins, as well as the interpolation weights corresponding to those events.

**Figure 4 sensors-25-03158-f004:**
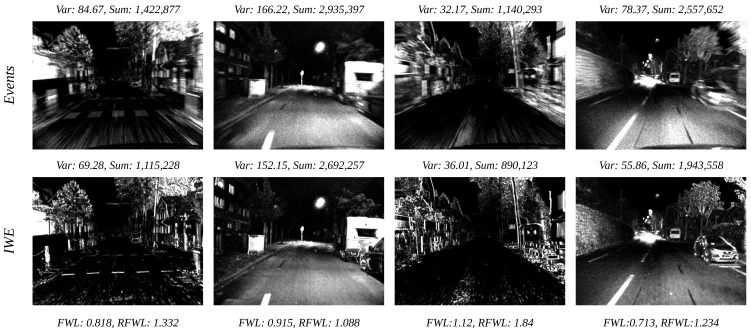
RFWL vs. FWL. The top row displays event count images from the original DSEC-Flow dataset [9], with the second row presenting the corresponding images of warped events (IWE). Variance and sum are indicated for each image. The bottom row shows the FWL and RFWL for the IWEs.

**Figure 5 sensors-25-03158-f005:**
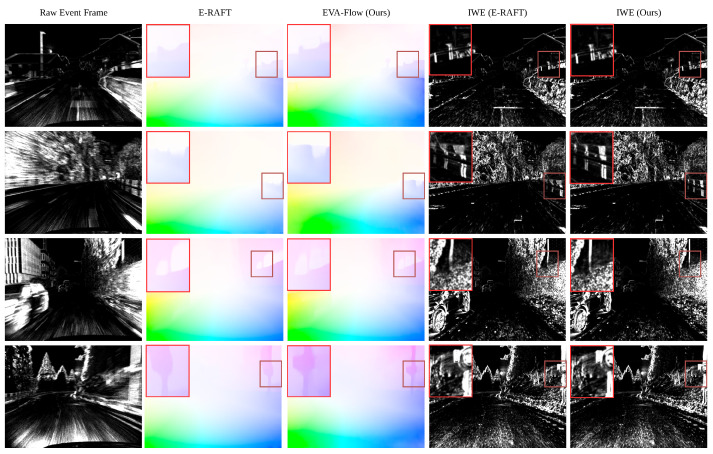
Qualitative examples of optical flow predictions on the DSEC-Flow test sequences. The four samples are taken from distinct scenes within different sequences. Regions of interest, marked with red boxes, are magnified 2× and shown in the top-left corner of each image.

**Figure 6 sensors-25-03158-f006:**
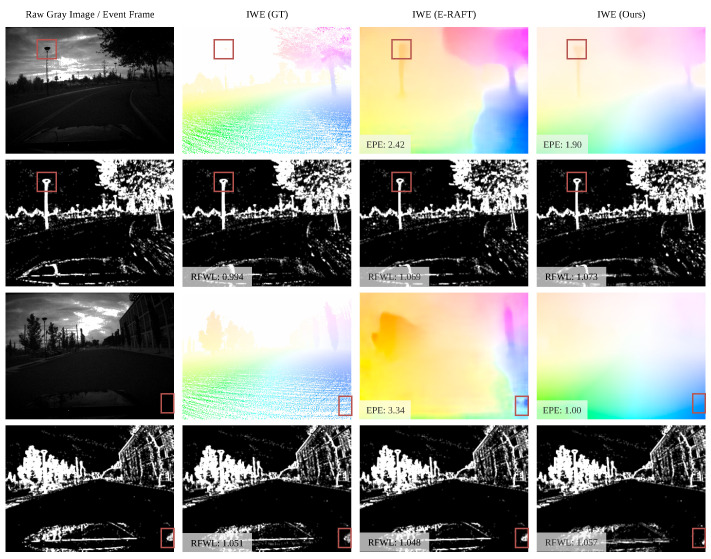
Qualitative examples of optical flow predictions on the outdoor day 1 sequence of the MVSEC dataset [2] (dt = 4). The model used here is only trained on the DSEC dataset [9]. The respective EPE and RFWL metrics can be found in the bottom left corner of the images. The areas of interest are denoted as red boxes in the images.

**Figure 7 sensors-25-03158-f007:**
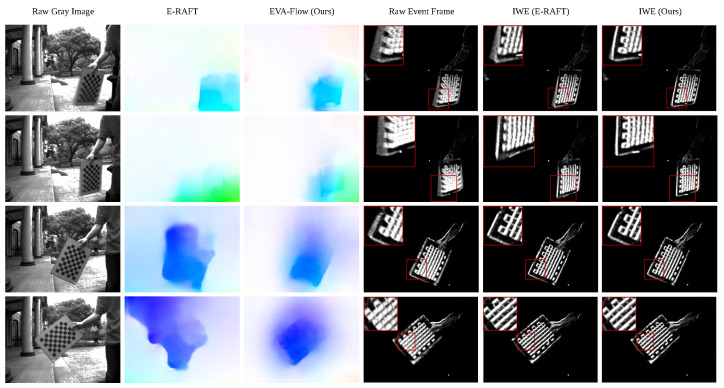
Qualitative examples of optical flow predictions on our EVA-FlowSet. The model used here is only trained on the DSEC dataset [9]. The initial two rows of samples are obtained from high-speed sequences, while the remaining two rows are taken from normal-speed ones, where red boxes highlight regions of interest (ROIs). Best view via zoom in.

**Figure 8 sensors-25-03158-f008:**
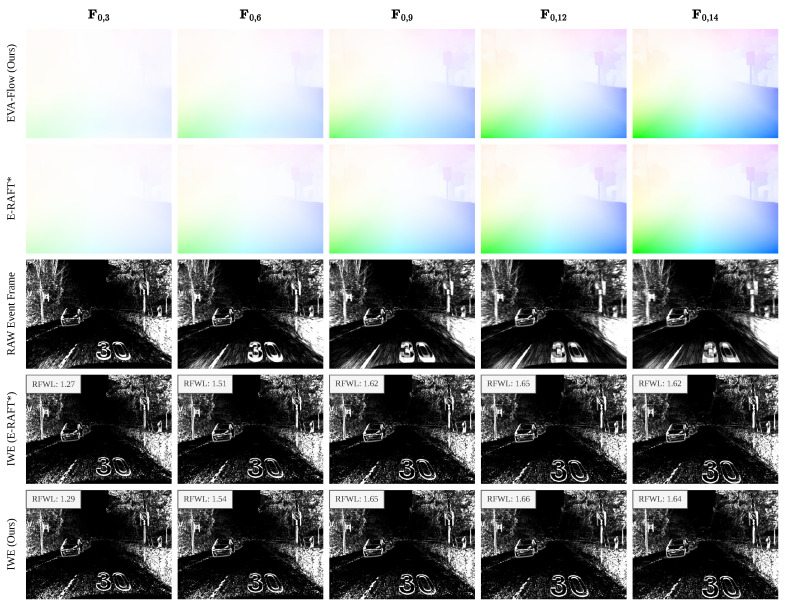
Illustration of time-dense optical flow evaluation. Each row presents results from different time steps within a 100 ms DSEC sample window (14 steps, #bins = 15). E-RAFT* denotes interpolated predictions across intermediate time steps.

**Figure 9 sensors-25-03158-f009:**
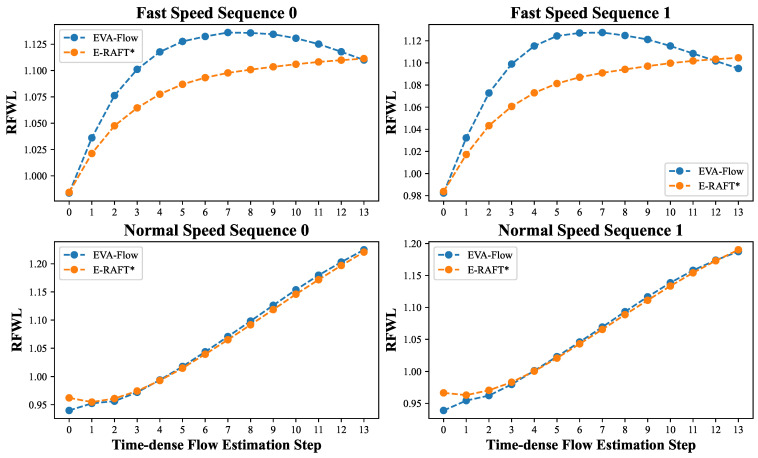
Evaluation results of time-dense optical flow using the RFWL in EVA-FlowSet. Models utilized here are exclusively trained on the DSEC dataset [9]. E-RAFT* denotes the time-dense optical flow obtained by interpolating the full-interval flow estimated by the original E-RAFT method.

**Figure 10 sensors-25-03158-f010:**
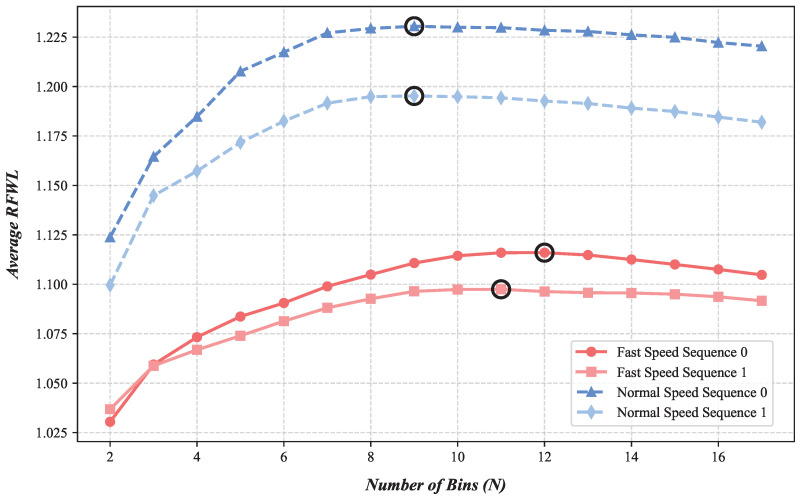
Impact of inference number of UVG bins (N) on time-dense flow quality (EVA-FlowSet, *T* = 20 ms), where black circles highlight the optimal metric points on each curve.

**Table 1 sensors-25-03158-t001:** Evaluation results on the DSEC-Flow’s test set [9].

Method	Params	GMACs	Latency	Prediction Rate	Supervision	EPE	AE	1PE	3PE
EV-FlowNet [2]	14 M	62	100 ms	10 Hz	**Self Supervised**	2.32	7.90	55.4	18.6
E-RFAT [3]	5.3 M	41	100 ms	10 Hz	10 Hz GT	0.79	2.85	12.7	2.7
IDNet 1iter [34]	**1.4 M**	55	**7.14 ms**	10 Hz	10 Hz GT	1.30	4.82	33.7	6.7
TIDNet [34]	1.9 M	55	**7.14 ms**	10 Hz	10 Hz GT	0.84	3.41	14.7	2.8
TMA [33]	6.9 M	56	100 ms	10 Hz	10 Hz GT	**0.74**	-	**10.9**	**2.3**
BFlow [39]	5.6 M	379	100 ms	**Bézier Curve**	10 Hz GT	0.75	2.68	11.9	2.44
Taming_CM [36]	-	-	10 ms	100 Hz	**Self Supervised**	2.33	10.56	68.3	17.8
LSTM-FlowNet [38]	53.6 M	444	10 ms	100 Hz	100 Hz GT	1.28	-	47.0	6.0
EVA-Flow (ours)	5.0 M	**16.8**	**5 ms**	**200 Hz**	10 Hz GT	**0.88**	**3.31**	**15.9**	**3.2**

**Bold**: best result, Underline: second best, Gray row: our method

**Table 2 sensors-25-03158-t002:** Evaluation results on MVSEC [2].

		dt = 1		dt = 4	
		EPE ↓	Outlier% ↓	EPE ↓	Outlier% ↓
SSL	EV-FlowNet [2]	0.49	0.20	1.23	7.30
	Spike-FlowNet [30]	0.49	-	1.09	-
	STE-FlowNet [44]	0.42	0.00	0.99	3.90
	Taming_CM [36]	0.27	0.05	-	-
USL	Hagenaars et al. [45]	0.47	0.25	1.69	12.5
	Zhu et al. [7]	0.32	0.00	1.30	9.70
	Shiba et al. [23]	0.30	0.10	1.25	9.21
SL	TMA [33]	0.25	0.07	**0.7**	**1.08**
	E-RAFT [3]	**0.24**	**0.00**	0.72	1.12
	EVA-Flow (Ours)	0.25	**0.00**	0.82	2.41
	E-RAFT [3] (Zero-Shot)	0.53	1.42	1.93	17.7
	EVA-Flow (Zero-Shot)	**0.39**	**0.07**	**0.96**	**4.92**

**Bold**: best result, Underline: second best, Gray row: our method

**Table 3 sensors-25-03158-t003:** Comparison of anytime flow estimation on DSEC-Flow.

Sequence	Method	RFWL														Avg.
		$F_{0\to 1}$	$F_{0\to 2}$	$F_{0\to 3}$	$F_{0\to 4}$	$F_{0\to 5}$	$F_{0\to 6}$	$F_{0\to 7}$	$F_{0\to 8}$	$F_{0\to 9}$	$F_{0\to 10}$	$F_{0\to 11}$	$F_{0\to 12}$	$F_{0\to 13}$	$F_{0\to 14}$	
thun_01_a	E-RAFT*	1.039	1.104	1.163	1.211	1.250	1.282	1.309	1.333	1.355	1.374	1.392	1.407	1.422	1.434	1.291
	EVA-Flow (Ours)	1.030	1.101	1.166	1.217	1.256	1.289	1.316	1.339	1.359	1.379	1.394	1.408	1.421	1.434	**1.294**
thun_01_b	E-RAFT*	1.058	1.134	1.193	1.236	1.277	1.313	1.349	1.382	1.416	1.447	1.476	1.500	1.522	1.539	1.346
	EVA-Flow (Ours)	1.046	1.135	1.208	1.262	1.305	1.340	1.370	1.393	1.419	1.444	1.467	1.490	1.512	1.532	**1.352**
interlaken_01_a	E-RAFT*	1.098	1.218	1.316	1.406	1.483	1.552	1.624	1.694	1.758	1.816	1.868	1.911	1.947	1.970	1.619
	EVA-Flow (Ours)	1.084	1.218	1.332	1.437	1.519	1.585	1.645	1.705	1.758	1.810	1.857	1.898	1.934	1.961	**1.625**
interlaken_00_b	E-RAFT*	1.105	1.233	1.338	1.426	1.500	1.565	1.621	1.670	1.714	1.753	1.788	1.819	1.843	1.860	**1.588**
	EVA-Flow (Ours)	1.088	1.229	1.340	1.437	1.515	1.576	1.626	1.670	1.709	1.742	1.770	1.796	1.816	1.837	1.582
zurich_city_12_a	E-RAFT*	1.005	1.020	1.034	1.050	1.065	1.079	1.090	1.103	1.116	1.127	1.139	1.149	1.161	1.169	**1.093**
	EVA-Flow (Ours)	1.004	1.021	1.032	1.046	1.060	1.075	1.085	1.099	1.111	1.122	1.134	1.145	1.157	1.166	1.090
zurich_city_14_c	E-RAFT*	1.057	1.155	1.250	1.329	1.391	1.453	1.510	1.555	1.598	1.636	1.666	1.695	1.724	1.752	1.484
	EVA-Flow (Ours)	1.044	1.152	1.249	1.333	1.394	1.460	1.517	1.561	1.605	1.645	1.675	1.704	1.732	1.761	**1.488**
zurich_city_15_a	E-RAFT*	1.071	1.174	1.256	1.324	1.384	1.436	1.480	1.522	1.566	1.606	1.642	1.676	1.703	1.721	1.469
	EVA-Flow (Ours)	1.062	1.177	1.270	1.345	1.409	1.461	1.502	1.542	1.580	1.614	1.646	1.677	1.701	1.720	**1.479**

E-RAFT*’s intermediate flow results from time-span interpolation of its predictions. **Bold**: best result.

**Table 4 sensors-25-03158-t004:** Experiments of different #bins on DSEC-Flow.

Model	#Bins (Training)	Test Sequences of 10 Hz				#Bins (Evaluating)	Validation Split of 5 Hz			
		EPE ↓	AE ↓	1PE ↓	3PE ↓		EPE ↓	1PE ↓	3PE ↓	5PE ↓
E-RAFT	15	0.79	2.85	12.7	2.7	15	2.96	43.5	17.6	10.3
EVA-Flow	6	0.955	3.29	16.7	3.9	11	**1.73**	**42.9**	**14.7**	**7.9**
	11	0.926	3.34	16.1	3.5	21	1.86	51.1	15.8	8.0
	15	0.895	3.39	16.1	3.3	29	1.82	49.4	15.6	8.0
	21	**0.877**	**3.31**	**15.9**	**3.2**	41	1.89	48.7	16.8	8.8
	31	0.901	3.37	17.0	3.2	61	2.02	53.5	18.2	9.4

**Bold**: best result, Underline: second best, Gray row: best of overall

**Table 5 sensors-25-03158-t005:** Ablations of event representation (#bins = 15).

Event Representation	AE ↓	EPE ↓	1PE ↓	3PE ↓
Voxel Grid [7]	3.48	0.96	17.7	3.65
Unified Voxel Grid	**3.39**	**0.89**	**16.1**	**3.30**

**Bold**: best result.

## Data Availability

The data presented in this study are openly available in https://github.com/Yaozhuwa/EVA-Flow (accessed on 15 May 2025).

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
