# Peer review of "Towards Anytime Optical Flow Estimation with Event Cameras"

_sensors, 2025, doi:10.3390/s25103158_

Round 1

Reviewer 1 Report

Comments and Suggestions for Authors

The main question addressed by the research is how to effectively estimate high frame rate optical flow from event-driven data, even in the absence of high frame rate ground truth. The authors tackle the limitations of current event-driven optical flow estimation methods by proposing the use of a Unified Voxel Grid for event representation and developing the EVA-Flow network to enable high frame rate event optical flows. Additionally, they introduce a novel Rectified Flow Warp Loss for evaluating intermediate optical flows. 
The event representation and loss function are the core innovations compared to previous work, and the authors have demonstrated the effectiveness of these improvements through experiments. The content of this manuscript has been well written and organized. It presents its own work well and compares it with existing methods, showing strong competitiveness. Here are some comments to help improve the article.
1. The four contributions can be streamlined by merging some of them, as there is some redundancy in the descriptions.
2. The authors gave an ablation study to the Unified Voxel Grid. However, the time interval used in Eq.(2) is not discussed. How sensitive is the algorithm to this parameter?
3. Minor problem: Unified Voxel Grid Representation (UVG). “Representation” does not need to be capitalized.

Reviewer 2 Report

Comments and Suggestions for Authors

Summary:

This paper proposes a deep learning framework called EVA-Flow for event-driven optical flow estimation. EVA-Flow outperforms existing event-driven optical flow methods in terms of latency, frame rate, and generalization, while maintaining competitive accuracy. The authors also collect and introduce a new real-world dataset, EVA-FlowSet, for evaluating the generalization ability of the model and temporal dense optical flow.

Strength:

(1) Event-driven optical flow estimation with low latency and high frame rate is achieved. EVA-Flow can generate optical flow output at a frame rate of up to 200Hz with a latency of only 5ms, which is significantly better than existing methods and solves the problem that existing methods are limited by the frame rate of the dataset.

(2) This paper introduces a spatiotemporal motion recurrence (SMR) module that can predict temporally dense optical flow and perform spatiotemporal refinement. The SMR module can generate temporally dense optical flow output and improve accuracy through spatiotemporal motion refinement.

(3) An unsupervised evaluation metric RFWL is proposed to evaluate the accuracy of event-driven optical flow. This metric can evaluate the accuracy of event-driven optical flow in an unsupervised manner, providing a basis for the reliability of time-intensive optical flow.

Weakness:

(1) Insufficient theoretical analysis.The theoretical advantages of UVG and SMR modules (such as mathematical proof of low latency) are not fully explained, and are only indirectly explained through experimental results. There is a lack of theoretical support for the convergence or stability of the algorithm.

(2) Limitations of comparative experiments. Without quantitative analysis of computational complexity or energy consumption, it is difficult to evaluate the applicability of the method in resource-constrained scenarios (e.g., drone or autonomous driving platform).

(3) Insufficient guidance for parameter adjustment. The impact of the hyperparameter N (the number of UVG bins) on performance is not systematically analyzed, and no parameter optimization suggestions for different scenarios (such as high-speed motion or low light) are provided, which may affect the practical application flexibility of the method.

(4) Lack of limitation analysis and possible improvement directions. An analysis of the limitations of the proposed method and possible future improvement directions should be included.

(5) Lack of related works. There are several other event-based optical flow works which cannot be introduced, such as [1], [2], et al. The authors may need to discuss the difference between them and the authors' work.

[1] Zhexiong Wan, et al. Learning Dense and Continuous Optical Flow from an Event Camera. TIP, 2022.

[2] Hanyu Zhou, et al. Exploring the Common Appearance-Boundary Adaptation for Nighttime Optical Flow. ICLR 2024.

Round 2

Reviewer 2 Report

Comments and Suggestions for Authors

The authors have addressed all my comments.